# Evidence Implicating Non-Dioxin-Like Congeners as the Key Mediators of Polychlorinated Biphenyl (PCB) Developmental Neurotoxicity

**DOI:** 10.3390/ijms21031013

**Published:** 2020-02-04

**Authors:** Carolyn Klocke, Pamela J. Lein

**Affiliations:** Department of Molecular Biosciences, University of California, Davis, School of Veterinary Medicine, Davis, CA 95616, USA; crklocke@ucdavis.edu

**Keywords:** arylhydrocarbon receptor (AhR), CREB signaling, dendritic arborization, developmental neurotoxicity, neurodevelopmental disorders, polychlorinated biphenyls, PCBs, ryanodine receptor (RyR), thyroid hormone receptor (THR)

## Abstract

Despite being banned from production for decades, polychlorinated biphenyls (PCBs) continue to pose a significant risk to human health. This is due to not only the continued release of legacy PCBs from PCB-containing equipment and materials manufactured prior to the ban on PCB production, but also the inadvertent production of PCBs as byproducts of contemporary pigment and dye production. Evidence from human and animal studies clearly identifies developmental neurotoxicity as a primary endpoint of concern associated with PCB exposures. However, the relative role(s) of specific PCB congeners in mediating the adverse effects of PCBs on the developing nervous system, and the mechanism(s) by which PCBs disrupt typical neurodevelopment remain outstanding questions. New questions are also emerging regarding the potential developmental neurotoxicity of lower chlorinated PCBs that were not present in the legacy commercial PCB mixtures, but constitute a significant proportion of contemporary human PCB exposures. Here, we review behavioral and mechanistic data obtained from experimental models as well as recent epidemiological studies that suggest the non-dioxin-like (NDL) PCBs are primarily responsible for the developmental neurotoxicity associated with PCBs. We also discuss emerging data demonstrating the potential for non-legacy, lower chlorinated PCBs to cause adverse neurodevelopmental outcomes. Molecular targets, the relevance of PCB interactions with these targets to neurodevelopmental disorders, and critical data gaps are addressed as well.

## 1. Introduction

Polychlorinated biphenyls (PCBs) are a structurally related class of 209 organochlorine compounds, individually referred to as congeners. Beginning in the late 1920s, PCB mixtures were mass-produced worldwide for diverse industrial and commercial applications, with the world production of PCBs estimated at 1.2–2 million tons, of which 0.2 to 0.4 million tons are believed to be “environmentally available” [1]. However, in 1979, prompted by evidence of their environmental persistence and growing concerns regarding human cancer risks, the United States banned commercial production of PCBs. This was followed by a global ban on PCB production instituted by signatory nations during the Stockholm Convention on Persistent Organic Pollutants (POPs) in 2001, which was appended in 2008 and 2014 [2,3]. These regulatory efforts resulted in the steady decline of environmental levels of legacy indicator PCBs (i.e., congeners predominant in the original commercial mixtures that are monitored as indicators of total PCB contamination). Yet, humans continue to be exposed to legacy PCBs because of their continued release from hazardous waste sites, PCB-containing equipment still in use, and construction materials used in buildings erected prior to the ban on PCB production [4,5,6]. Background levels of PCBs in environmental media are reported to range from 1–100 pg/m^3^ in air and 100–1000 pg/g dry weight in soil [7]. In addition, data emerging over the past decade demonstrates widespread human exposure to non-legacy PCB congeners that were not present in the commercial PCB mixtures [8]. These non-legacy PCBs are detected in indoor and outdoor environments and in human tissues and the most likely source of these contaminants is off gassing from common household paints [9,10]. More than 50 non-legacy PCBs have been detected in paint pigments as inadvertent byproducts of industrial pigment and dye production at levels ranging from 2–200 ng/g fresh weight [11]. Airborne PCB emissions of these non-legacy PCBs can exceed 500 pg/m^3^ in a recently painted room [12]. Of concern, non-legacy PCBs have also been detected in the plasma of pregnant women at levels ranging from 0.005 ng/mL to 1.717 ng/mL [13]. In contrast to the commercially produced PCBs, levels of the non-legacy PCBs are increasing in the environment and in human tissues [7,11].

Both human and animal studies identify the developing brain as a vulnerable target of PCBs. Multiple reviews of the epidemiologic literature have concluded that exposure to PCBs during critical developmental periods increases the risks of neuropsychological deficits in children, demonstrated by impaired executive and psychomotor function, as well as deficits in attention, learning, and memory [14,15,16,17]. More recent studies suggest that developmental PCB exposures may also increase risk of neurodevelopmental disorders (NDDs) [14,15], specifically, autism spectrum disorders (ASD) [18,19,20,21,22,23] and attention-deficit/hyperactivity disorder (ADHD) [24,25,26,27], and they confirm the association between early life exposure to PCBs and general cognitive impairments without NDD diagnosis [17,28,29,30,31]. Prenatal exposure to PCBs is also associated with increased risk of low birth weight, defined as < 2500 g at birth) [32,33,34,35,36] and small for gestational age [37,38,39], both of which are prognostic indicators of poor neurological outcome [40,41,42,43]. However, even after adjusting for birth weight and size, the positive association between developmental PCB exposures and neuropsychological deficits remains. Experimental animal studies confirm that developmental PCB exposures cause neurobehavioral effects similar to those observed in humans and these outcomes occur in the absence of adverse effects on reproductive and birth outcomes [44,45,46,47]. However, which of the 209 PCB congeners contributes to PCB developmental neurotoxicity, and the mechanism(s) by which PCBs interfere with neurodevelopment, remain critical questions. This review focuses on recent research that is beginning to shed light on these questions.

## 2. PCB Nomenclature and Classifications

Chemically, PCBs are a complex mixture of chlorinated biphenyl isomers, or congeners, that differ in structure. Individual congeners are identified by a numerical designation from 1 to 209, with ascending numbers indicating increasing chlorine substituents on the biphenyl backbone [48]. Congeners with four or less chlorine substituents are referred to as lower chlorinated PCBs (LC-PCBs); those with more than four chlorine substituents, as higher chlorinated PCBs (HC-PCBs) [8]. PCBs can be metabolically hydroxylated or sulfonated, and 19 congeners are stable atropisomers or enantiomers, with chiral asymmetry about their biphenyl bond axes [8]. Theoretically, each PCB enantiomer and metabolite may have distinct interactions with biological targets and this diversity of structures and potential modes of action explains, in part, why PCB developmental neurotoxicity has been a complex research issue.

PCB congeners can also be subdivided into semi-volatile versus relatively non-volatile congeners. Volatility is influenced by the number of chlorine substituents, with the HC-PCBs typically being less volatile. Both LC-PCBs and HC-PCBs were constituents of the legacy commercial PCB mixtures, and both LC-PCBs and HC-PCBs are present in paint pigments; however, many of the non-legacy PCBs associated with paint pigments are LC-PCBs [8]. The HC-PCBs are more likely to bioaccumulate and biomagnify up food chains, whereas the LC-PCBs comprise the majority of airborne PCBs found in major cities in the United States (U.S.) and in indoor air in U.S. schools (reviewed in [8]). Based on the differential environmental distribution of HC-PCBs vs. LC-PCBs, it is proposed that humans are primarily exposed to HC-PCBs via diet, particularly fish, and to LC-PCBs via inhalation [8]. However, it is worth noting that recent studies have documented the widespread presence of the LC-PCB congener, PCB 11, in commercial milk products in Northern California [13]; and, conversely, the HC-PCB congener, PCB 95, is the second most abundant PCB detected in the air in U.S. schools [49].

PCBs can also be classified according to their planar structure, which is determined by the positioning of chlorine substituents around the biphenyl backbone. PCBs with no chlorines in the *ortho* position assume a coplanar geometry of the rings, and PCBs with one to four chlorines in the *ortho* position assume increasing degrees of noncoplanar ring geometry (Figure 1). Coplanar congeners can bind the aryl hydrocarbon receptor (AhR), an intracellular ligand-activated basic helix-loop-helix/Per-Arnt-Sim (bHLH/PAS) transcription factor that is the canonical receptor for 2,3,7,8-tetrachlorodibenzo-*p*-dioxin (TCDD). Coplanar PCBs that bind the AhR are classified as dioxin-like (DL) PCBs [50,51,52]. Of the 209 PCB congeners, twelve (PCBs 77, 81, 105, 114, 118, 123, 126, 156, 157, 167, 169, and 189) are identified as DL-PCBs [53]. In contrast, noncoplanar congeners have little to no AhR binding affinity, and thus are referred to as non-dioxin-like (NDL) PCBs [50,51,52]. Importantly, NDL PCBs represent a significantly greater percentage of the PCBs detected in human serum, adipose tissue, breast milk, and brain tissue from children diagnosed with a NDD [19,54,55,56].

Although environmentally relevant exposures to dioxin and DL PCBs are associated with adverse outcomes in several organ systems, especially skin, liver and the immune system [57,58,59], and some are probably carcinogenic [60], there is little data demonstrating that DL PCBs are direct developmental neurotoxicants (but see [29,61,62]). In contrast, human, animal and mechanistic studies confirm the developmental neurotoxicity of legacy NDL PCBs [14,19,21,26,31,63,64,65,66], and emerging evidence suggests that non-legacy LC-PCBs also pose a risk to the developing brain [67].

## 3. Neurobehavioral Studies of PCB Developmental Neurotoxicity

There is an extensive body of literature describing the developmental neurotoxicity of legacy PCBs in experimental animal models, which has been previously reviewed [44,45,46,68]. Many of these studies, particularly the earlier studies, focused on the neurobehavioral effects of developmental exposures to the legacy commercial mixtures, notably the Aroclor mixtures. Here, we will focus on reviewing those studies that have been published in the past decade, several of which investigated the developmental neurotoxicity of individual PCB congeners. A key consideration in evaluating these studies is the relevance of the exposure paradigm to the human condition, both in terms of the route of exposure and the dose. A range of PCB doses and routes of administration have been employed in animal studies, and, in most cases, PCB levels were not measured in exposed animals. To provide a point of reference for this section, the PCB body burden in contemporary humans is reported to be about 2.2 ng/mL in maternal sera [69] and between 0.7 to 66 ng/g wet weight in postmortem brain tissue from individuals across the U.S. and Europe [56,70,71,72]. PCB levels in the brain tissue of juvenile rats exposed to Aroclor 1254 at 1 mg/kg/day throughout gestation and lactation via the maternal diet ranged from 0.5 to 3.0 ng/g wet weight [47]. These data suggest that, with some exceptions, the PCB doses used in many of the animal studies discussed below result in tissue levels that are within the range observed in humans. Therefore, it is reasonable to conclude that behavioral deficits occurring from such exposures are relevant to informing human health risks.

While a diverse number of neurobehavioral outcomes have been assessed in animal models of developmental PCB exposure, for the purposes of this review, we will focus on behavioral domains of relevance to NDDs, specifically: (1) altered locomotor activity (hyperactivity or hyperkinetic behavior) and attention; (2) deficits in social interaction and communication; and (3) impaired cognitive and executive function. Below, we summarize the recent animal literature published within the past decade that report effects of developmental PCB exposure on these behavioral domains.

### 3.1. Hyperactivity

Hyperactivity, or hyperkinetic behavior, is a hallmark symptom of ADHD, and deficits in sustained attention are present in both ADHD and ASD [73]. The tests for hyperactivity in experimental animal models are typically simple evaluations of locomotor behavior, such as spontaneous locomotor activity or open field, or are derived from other quantitative measurements of animal movement (e.g., distance traveled across an arena) in other tests. Table 1 summarizes the recent animal literature evaluating locomotor behavior following developmental PCB exposures.

Effects of developmental PCB exposure on locomotor activity are reported to be sex-specific and to depend on the degree of chlorination in the PCB congener. In mice, orogastric exposure to Aroclor 1254 at 6 or 18 mg/kg/d during lactation and the juvenile period significantly enhanced locomotor activity in females but not males, evidenced as increased travel distance and time in the center of an open-field arena [74]. Similarly, gestational and lactational exposure to Aroclor 1254 at 18 mg/kg/d (i.p.) resulted in increased locomotor activity in an open-field arena, with significant enhancement of total distance traveled and distance within the center of the arena [75]. However, sex was not specified in this study. In contrast, another study evaluated lactational exposure to a mixture of NDL PCBs (28, 52, 101, 138, 153, and 180) via the orogastric route in mice at 1, 10, or 100 ng/kg/d and observed a decrease in total locomotion in male offspring only [76].

In rats, exposure to Aroclor 1221 at 0.5 or 1.0 mg/kg/d (i.p.) on gestational days (GDs) 16 and 18 decreased locomotion in male offspring, as indicated by reduced distance travelled in the light-dark box (LDB) test [77]. While Aroclors are comprised predominantly of NDL congeners, they do contain DL PCBs [80], so it is not possible to tell which congener type(s) mediated the effects. Further studies in rats evaluating gestational and lactational exposure via the maternal diet to single NDL congeners (PCB 52, 138 or 180) at 1 mg/kg/d revealed differential effects of each congener on locomotor activity between males and females. PCB 52 was found to have no effect on motor activity in either sex, while PCB 138 decreased activity in both males and females and PCB 180 reduced activity in males, but not females [78]. In a second study by the same group using the same exposure paradigm, motor coordination was assessed using the rotarod test. In this study, developmental exposure to PCB 52, but not PCBs 138 or 180, decreased latency to fall from the rotarod apparatus [79].

These data suggest that developmental exposure to NDL PCBs decreases locomotor activity in a congener-dependent, and, in some cases, sex-specific manner. These observations do not support a role for NDL PCBs as risk factors for ADHD, at least not for the hyperactivity component of the disorder. However, this is an extremely limited data set, and experiments in which animals were developmentally exposed to Aroclor mixtures suggest that developmental PCB exposure increases locomotor activity. Several potential explanations for this discrepancy include: (1) Aroclor mixtures contain a subset of NDL PCBs that exert effects on locomotor behavior that are different than those caused by the individual NDL congeners that have been tested for locomotor effects; (2) DL PCBs in the Aroclor mixtures increase locomotor activity, and this effect predominates over the tendency of NDL PCBs to decrease locomotor activity; and/or (3) Complex interactions between NDL and DL congeners in Aroclor mixtures influence locomotor outcomes (see Section 5.1). Distinguishing between these possibilities will require structure activity relationship (SAR) studies of the dose-related effects of PCBs on locomotor behavior.

### 3.2. Social Deficits

Impaired recognition of social cues, reduced social interaction, and decreased communication are clinical features of ASD, but do not present as frequently in ADHD [81]. Evaluating social behavior in rodents and other experimental animal models is challenging because not all aspects of social interaction between humans are recapitulated in rodents [82]. However, rodent models can capture core aspects of sociability [83]. Broadly, sociability in rodents is defined as an individual animal’s preference for investigating and spending time with another conspecific. Sociability can be directly tested in a 3-chamber social approach assay and similar related assays [82,84,85,86]. Ultrasonic vocalizations (USVs) and sociosexual choice are other frequently used tasks to assess sociability and communication in rodent models [87,88]. Table 2 summarizes the recent animal literature evaluating the effects of developmental PCB exposure on social behavior (earlier studies have been previously reviewed [44]).

A recent study using a mouse model evaluated the effects of prenatal and lactational exposure to a mixture of six NDL indicator PCB congeners via the maternal diet at 10 or 1000 ng/kg/d. This exposure enhanced sociability and social approach in both male and female offspring, but reduced nose-nose sniff interactions in male offspring [89]. Studies using rat models provide stronger support for the hypothesis that developmental PCB exposure causes social behavior deficits. Exposure of pregnant female rats to Aroclor 1221 at 0.5 or 1.0 mg/kg/d (i.p.) on GD 16 and 18 reduced social interaction with a novel conspecific in male adult offspring [91]. Interestingly, this effect was seen at the lower dose (0.5 mg/kg/d) and not the higher dose (1.0 mg/kg/d) of Aroclor 1221. In a second study from this group using the same exposure paradigm, USV calls in a sociosexual context (i.e., in the presence of a conspecific of the opposite sex) increased but nose-nose sniff interactions decreased in adult offspring [92]. A separate rat study that also evaluated Aroclor 1221, but used a different exposure paradigm (1 mg/kg i.p. on GD 16, 18, and 20 and on PNDs 24, 26, and 28), observed increased USV calls and affiliative wrestling behavior in female offspring and reduced social novelty preference in male offspring [90].

To date, animal studies of the effects of developmental PCB exposure on social behavior have largely employed complex mixtures. One exception is a recent study using a rat model that explored the effects of gestational and lactational exposure via the maternal diet to two higher doses (12.5 mg/kg/d and 25 mg/kg/d) of a combination of the NDL PCB 47 and the DL PCB 77. This exposure paradigm significantly impaired social recognition and social investigation behaviors [93]. Unfortunately, the authors of this study did not investigate the social effects of the individual PCB congeners. Overall, developmental PCB exposure tended to decrease sociability in males, which is consistent with the sex bias of ASD in humans. However, based on the available data, it is not possible to discern whether these effects are attributable to a specific class of PCBs.

### 3.3. Cognitive Impairment and Executive Dysfunction

Cognitive function refers to the ability of an organism to process information in their environment, and includes learning, memory, and attention [94]. Executive functions fall under cognitive function, and are defined as higher-order processes involved in the planning and control of goal-oriented behavior, including the initiation, inhibition, and shifting of behavioral responses [45]. Both cognitive and executive functions are affected in NDDs to varying degrees, though the clinical presentation of alterations in these domains are heterogeneous in humans. A review of earlier human literature of PCB developmental neurotoxicity concluded that executive function is possibly the most impacted behavioral domain associated with developmental PCB exposure, with working memory and inhibitory control appearing to be the most affected [45].

A hallmark symptom of ADHD is impulsivity, which is broadly defined as acting or behaving without first considering potential outcomes [95]. Impulsivity, or lack of inhibitory control, can be tested in experimental animal models using various operant paradigms, such as reversal learning (RL), differential reinforcement of low rate (DRL), and fixed interval (FI). Additional aspects of cognitive function, such as learning and memory, can also be probed using these assays. Spatial learning and memory are commonly assessed using the Morris water maze (MWM) and delayed spatial alternation (DSA), while novel object recognition (NOR) evaluates object-based attention and short-term memory [96]. Table 3 summarizes the recent animal literature evaluating the effects of developmental PCB exposure on cognitive behavior and executive function.

In mice, direct exposure to Aroclor 1254 at 6 or 8 mg/kg/d via gavage from PND 7–42 reduced short-term memory in female, but not male, offspring tested in the NOR task [74], but had no effect on spontaneous alternation behavior in either sex. Another mouse study found that i.p. administration of Aroclor 1254 at 18 mg/kg/d to the dam throughout gestation and lactation decreased object-based attention, evident as reduced object recognition in the NOR task and increased impulsivity in an EPM task in offspring [75]. In contrast, mice exposed to a mixture of six indicator NDL PCB congeners at 1, 10, or 100 ng/kg/d via the maternal diet during the lactational period exhibited no deficits in spatial learning and memory in an MWM task or the spontaneous alternation task across two separate studies [76,97]. The lack of effect in the study of the NDL PCB mixture compared to the previously discussed studies of Aroclor mixture effects on cognitive function, likely reflects the 2 to 3 orders of magnitude lower dose levels used in the NDL PCB studies. In contrast, developmental exposure to this low level NDL PCB mixture increased anxiety-like behavior in EPM and LDB tasks [76].

The effects of PCB mixtures on cognitive and executive function have also been studied in rat models. Gestational and postnatal exposure to 1 mg/kg/d (i.p.) Aroclor 1221 increased the number of entries into and time spent within the open arms of the EPM in female but not male offspring [90]. Several rat studies have used a PCB mixture custom made to proportionally mimic the PCB congener profile in fish caught in the Fox River in Wisconsin [100]. Rats were exposed to this Fox River mixture, which was comprised of Aroclors 1242 (35%), 1248 (35%), 1254 (15%) and 1260 (15%), at 3 or 6 mg/kg/d throughout gestation and lactation via the maternal diet. This exposure paradigm produced female-specific deficits in the DRL test and impaired inhibitory control in both sexes [98]. Interestingly, the female DRL deficits were only observed in the low (3 mg/kg/d), but not the high (6 mg/kg/d) dose group. Similar non-monotonic dose-response effects on cognitive behavior were reported in an older study of rats exposed developmentally to Aroclor 1254 in the maternal diet [47]. Postnatal exposure to the Fox River PCB mixture produced a different behavioral outcome in which males in the 6 mg/kg/d group had fewer errors and decreased perseverative responses in an attentional set-shifting task, which measures cognitive flexibility [99]. Despite making fewer response errors, these animals had increased latency to respond in the set-shifting task. There appeared to be no significant effect of PCB exposure on response inhibition in a DRL task in either sex. The differential effects of developmental PCB exposure across these two studies underscores the importance of timing of exposure.

The neurobehavioral effects of developmental exposure to purified NDL PCB congeners in the absence of DL PCBs have been assessed in a rat model. Gestational and lactational exposure to PCBs 52, 138, or 180 at 1 mg/kg/d in the maternal diet caused learning deficits in a Y maze task in which the animals had to associate wall color with the location of a food reward [79]. Both male and female offspring exposed to PCBs 138 or 180, but not PCB 52, required more trials to reach the learning criterion of 10 correct responses per day.

Collectively, the neurobehavioral studies published over the past decade extend the older animal literature documenting cognitive deficits in animal models of developmental PCB exposure (reviewed in [45,46]). They also provide evidence to suggest that environmentally relevant exposures to NDL PCBs phenocopy the effects of Aroclor mixtures on cognitive behavior. This latter observation is consistent with evidence from earlier rodent studies suggesting that NDL PCBs [63,64,101] and not DL PCBs [65] are the primary driver of many of the PCB-associated cognitive and behavioral abnormalities observed in humans. Nonetheless, this summary of the recent animal literature clearly demonstrates the need for more comprehensive SAR studies of the neurobehavioral effects of developmental exposures to specific PCB congeners at environmentally relevant levels in order to identify which PCB congeners are developmental neurotoxicants and to determine whether the profile of developmental neurotoxicity varies in a congener-specific manner.

## 4. Neurodevelopmental Processes Altered by PCBs

Key to elucidating mechanism(s) of PCB developmental neurotoxicity is the identification of specific neurodevelopmental processes that are altered by PCBs. A recent review of the human and animal literature with respect to the neuropathology of PCB developmental neurotoxicity concluded that even at levels that are overtly toxic, PCBs do not cause major structural changes in the developing brain [14]. Rather, it appears that the neurobehavioral deficits linked to developmental PCB exposures are due to subtle organizational defects in neurodevelopment that give rise to altered patterns of synaptic connectivity [14,102,103,104]. The synaptic patterns established in the developing brain critically influence cognitive function later in life [105], and altered patterns of synaptic connectivity are associated with NDDs and cognitive impairment [106,107,108,109,110,111,112]. Two neurodevelopmental processes that are important determinants of synaptic connectivity—neuronal apoptosis and the morphogenesis of axons and dendrites—have been shown by more than one laboratory to be altered by PCBs.

### 4.1. PCB Effects on Neuronal Apoptosis

Neuronal apoptosis is essential to typical brain development [113,114,115], occurring in proliferative zones and in postmitotic neurons in the neonatal brain [116]. During development, neuronal apoptosis is under tight spatiotemporal control. Disruption of either the timing or the magnitude of apoptosis in a given brain region can affect the total cell number and, consequently, neuronal connectivity, resulting in deficits in higher-order function even in the absence of obvious pathology [114,115,117,118]. Histologic studies in a rat model of PCB developmental neurotoxicity indicate that exposure to Aroclor 1254 at 0.1 or 1.0 mg/kg/d in the maternal diet throughout gestation and lactation significantly increases apoptosis in the brain of offspring at PND 1 but not PND 21 [119]. Specifically, assays of caspase-3 activity, as well as terminal deoxynucleotidyl transferase dUTP nick-end labeling (TUNEL) staining of the cortex, hippocampus, and cerebellum, revealed significantly enhanced apoptosis in all three brain regions, with the most pronounced increase in the cerebellum [119].

In vitro studies confirm that Aroclor 1254, and other Aroclor mixtures, induce apoptosis in primary neuronal cell cultures at concentrations that do not cause significant cytotoxicity [120,121]. Studies of individual PCB congeners suggest that the pro-apoptotic activity of these complex PCB mixtures are largely mediated by NDL PCB congeners. An investigation of the potential apoptotic activity of PCB 52 in human neuronal SK-N-MC cells found that this NDL congener triggered apoptosis via p53-independent mechanism(s) [122]. In a comparative study of PCB 47, an NDL congener, and PCB 77, a DL congener, in primary rat hippocampal neurons, PCB 47, but not PCB 77, was observed to significantly increase caspase-dependent apoptosis [121]. Neither PCB 47 nor PCB 77 triggered neuronal apoptosis in primary rat cortical neurons established from the same dissection and grown under the same culture conditions. Another comparative study of NDL vs. DL PCB congeners using primary cultures of rat cortical neurons indicated that both the NDL congener, PCB 153, and the DL congener, PCB 77, accelerated apoptosis in a time- and concentration-dependent manner [123]. The extent of apoptosis was greater in the cultures treated with PCB 77 compared to the cultures treated with PCB 153. The reason(s) for the different outcomes between the two studies with regards to the effects of PCB 77 on neuronal apoptosis may reflect subtle differences in culture systems: primary rat cortical neurons were maintained under serum-free conditions in one study [121] but in the presence of serum in the other [123]. However, a more likely explanation is the significant differences in the concentrations of PCB 77 that were tested. The study in which PCB 77 was observed to have no effect on neuronal apoptosis tested PCB 77 at concentrations ≤ 1.0 µM because cytotoxic effects were observed at concentrations > 1.0 µM [121]. In contrast, in the study that observed PCB 77-induced neuronal apoptosis, PCB 77 was tested at 30, 50, and 100 µM, concentrations that caused significant cytotoxicity within 1–3 h after exposure [123]. The relevance of neuronal apoptosis caused by exposure to overtly cytotoxic levels of PCBs is questionable. Interestingly, the group that reported PCB 47-induced apoptosis in primary hippocampal neurons, also observed that the NDL congener PCB 104 had no effect on baseline apoptosis in either primary rat hippocampal or cortical neurons [121]. The reason(s) for the differential effect of PCB 47 versus PCB 104 on neuronal apoptosis likely reflects differences in their relative activity at the ryanodine receptor (RyR), as discussed in Section 5.3. Importantly, this observation demonstrates that pro-apoptotic activity is not an inherent property of all NDL PCBs.

### 4.2. PCB Effects on Axonal and Dendritic Morphogenesis

Axonal and dendritic morphology are principal determinants of synaptic architecture [124,125]. The size of the dendritic arbor determines the total synaptic input a neuron can receive, and both axonal and dendritic branching patterns influence the types and distribution of these inputs [126,127,128,129]. Moreover, dendritic shape is refined by experience, and dendritic structural plasticity is considered the cellular substrate of learning and memory [130]. Abnormalities in neuronal structure, i.e., axonal and dendritic number, length, and branching, are thought to underlie the clinical symptomologies of human NDDs [81,105,131,132,133,134], and in experimental animal models, even subtle perturbations of spatial or temporal aspects of axonal or dendritic growth are associated with altered neurobehavior. For example, rodent models demonstrate that disrupted axonal growth [135,136] or delays in dendritic maturation of the neocortex as a result of transient depletion of cholinergic input [137,138] can cause persistent learning and memory deficits. Increased rates of dendritic growth, as observed in animals exposed to cocaine in utero, are also associated with cognitive impairment [139,140].

At least one study has examined the effects of developmental PCB exposure on axonal outgrowth in vivo [141]. In this study, pregnant rats were exposed to Aroclor 1254 in the diet at 125 ppm throughout gestation and lactation. Weaned offspring were maintained on chow containing 125 ppm Aroclor 1254 until brains were collected for Timm’s silver sulfide staining. The authors observed that developmental exposure to Aroclor 1254 significantly reduced the relative size of II-P mossy fibers in 16-, 30- and 60-day old rats. In contrast, the exposure had no effect on the size of hilar or suprapyramidal mossy fibers or on cortical thickness. The reason(s) for the selective sensitivity to Aroclor 1254 of granule cells that give rise to II-P mossy fibers remains unknown.

Three studies have published observations of in vivo effects of developmental exposures to Aroclor 1254 on dendritic morphogenesis in three independent cohorts of rats of the same strain (Long-Evans) [47,142,143]. In all three studies, dams were exposed to Aroclor 1254 in the diet preconception and continuing throughout gestation and lactation, and all three studies tested Aroclor at 6 mg/kg/d, although one study [47] also tested a lower dose of 1 mg/kg/d. All three groups used Golgi staining to quantify dendritic arborization of individual neurons in the offspring’s brain. Two of the three studies reported that developmental PCB exposure significantly altered dendritic arborization [47,142]. The first of these two studies [142], which quantified dendritic arborization only in male offspring, observed that Aroclor 1254 at 6 mg/kg/d caused a pronounced age-related increase in the rate of dendritic growth in CA1 hippocampal pyramidal neurons and cerebellar Purkinje cells. While dendritic lengths were significantly attenuated in PCB-exposed animals at PND 22, dendritic growth was comparable to or exceeded that observed in vehicle controls at PND 60 [142]. The second positive study [47], which measured dendritic growth at PND 31 in male offspring, reported that developmental exposure to Aroclor 1254 increased basal dendritic arborization but blocked experience-dependent dendritic growth in cerebellar Purkinje cells and neocortical pyramidal neurons. In the cerebellum, these dendritic effects were only observed in the 1 mg/kg/d exposure group, and in the neocortex, dendritic effects were more pronounced in the 1 mg/kg/d exposure group compared to the 6 mg/kg/d exposure group. Interestingly, the lower, but not the higher, dose of Aroclor 1254 was associated with significant deficits in spatial learning and memory in the MWM [47]. In contrast, the third study [143], which quantified the dendritic arbor of Purkinje cells at PND 21, reported no changes in primary dendrite length or branching area. This observation is consistent with the second study [47] that also observed no effects on dendritic arborization in Purkinje cells in the 6 mg/kg/d Aroclor 1254 exposure group. However, both the second and third studies are at odds with the first study [142], which reported that developmental exposure to Aroclor 1254 at 6 mg/kg/d significantly altered the dendritic arborization of Purkinje cells. The reason(s) for this discrepancy between studies are not known, but since it appears that developmental PCB exposure alters the rate of dendritic growth, perhaps the differences between studies reflect the fact that dendritic arborization was quantified at different ages.

Studies of Aroclor 1254 do not provide insights as to which PCB congener(s) influence dendritic arborization during development; however, a subsequent study [144] demonstrated that developmental exposure to PCB 95, a NDL congener, phenocopies the dendritic effects of Aroclor 1254. In this study, rats were exposed throughout gestation and lactation to PCB 95 at 0.1, 1, or 6 mg/kg via the maternal diet and the morphology of hippocampal CA1 pyramidal neuronal dendrites was assessed at PND 38 using Golgi staining. Dendritic arborization was significantly increased in pups exposed developmentally to PCB 95 at 0.1 or 1 mg/kg/d. However, dendritic growth in the hippocampus in the 6 mg/kg/d exposure group was not significantly different from that of vehicle controls, recapitulating the non-monotonic dose-response relationship observed earlier for Aroclor 1254 effects on dendritic arborization. In vitro studies confirm that PCB 95, and another NDL congener, PCB 136, promote dendritic growth (reviewed in [14]). At pM to nM concentrations, PCB 95 [47,144,145,146] and PCB 136 [147], increase dendritic arborization in primary rat hippocampal and cortical neuron grown in neuron-glia co-cultures at high cell density. These observations suggest that the effects of PCBs on dendritic growth are autonomous to the brain and occur independent of systemic effects of PCBs. Interestingly, the morphogenic effect of these NDL PCBs is selective to dendritic growth in that axonal growth was not altered in PCB-exposed cultures relative to vehicle controls. The dendritic phenotype exhibits a non-monotonic concentration–response relationship with enhanced dendritic growth observed at nM to pM concentrations but not at low µM concentrations [144,147]. The biological explanation for the lack of response at the higher end of the concentration response curve remains to be answered, but does not reflect decreased cell viability [144,147]. The dendrite-promoting effects of PCB 95 have also been demonstrated in primary mouse hippocampal and cortical neurons, although in this species, the effect is sex-specific, with the sex-specificity varying between the two brain regions [148]. The species and regional differences are thought to reflect species and regional differences in the rates of neuronal maturation [148]. PCB 95 also promotes the formation of dendritic spines in primary rat hippocampal neurons [149].

Studies from other laboratories have shown that the hydroxylated metabolites of NDL PCBs can promote dendritic growth in vitro. The 4-hydroxy metabolites of PCBs 112, 165, and 187 enhanced dendritic arborization in primary mouse cerebellar Purkinje cells [150]. Interestingly, this same group demonstrated that the hydroxylated metabolites of NDL PCBs 106, 121, and 159 had no dendrite-promoting activity in this same culture system; however, these compounds inhibited thyroid hormone (TH)-induced dendritic growth [150,151]. These observations suggest that dendrite-promoting activity is not a property of all NDL PCBs. Consistent with this observation, another laboratory reported that PCB 66, a NDL congener whose physicochemical properties are very similar to those of PCB 95, has no effect on dendritic arborization in primary rat hippocampal neurons when tested at the same concentration range over which PCB 95 increases dendritic growth [144]. The properties that confer dendrite-promoting activity may be linked to the ability to sensitize the RyR, as discussed in more detail in Section 5.3 (and reviewed in [14,55]).

Whether DL PCBs influence dendritic arborization in the developing brain is not known. It has been reported that mice born to dams administered TCDD at 0.6 or 3 μg/kg on GD 12.5 exhibit increased growth of dendritic branches in both the hippocampus and amygdala at PND 14, and significantly reduced dendritic spine densities at 16 months of age [152]. These observations suggest the possibility that DL PCBs may modulate dendritic growth in the developing brain, but this has yet to be directly tested. A more comprehensive understanding of the SAR of the dendrite-promoting activity of PCBs remains a critical data gap.

## 5. Molecular Mechanisms of PCB Developmental Neurotoxicity

The most widely postulated mechanisms of PCB developmental neurotoxicity include: (1) disruption of the hypothalamic-pituitary-thyroid axis (reviewed in [153,154]); (2) altered neurotransmitter signaling, specifically decreased dopamine levels (reviewed in [14,155]) and potentiated GABA signaling (reviewed in [67]); (3) modulation of neuronal Ca^2+^ signals (reviewed in [55]); and (4) increased intracellular levels of reactive oxygen species (ROS) (reviewed in [155,156]). Both NDL and DL PCBs have been shown to interfere with TH homeostasis (see Section 5.2) and to increase intracellular ROS levels [155,156]. In contrast, comparative studies of NDL versus DL congeners indicate that NDL PCBs, but not DL PCBs, deplete intracellular dopamine concentrations [157,158,159], potentiate GABA signaling [160,161,162], and alter Ca^2+^-dependent signaling [163,164,165].

Each of the four biological activities associated with NDL PCBs has been implicated in the control of neuronal apoptosis and/or neuronal morphogenesis (axonal and/or dendritic growth) during typical neurodevelopment. However, to date, experimental evidence supporting a causal role for these biological activities in PCB effects on these neurodevelopmental processes has been reported only for Ca^2+^ and ROS; therefore, this section will address these two biological activities as mechanisms contributing to PCB developmental neurotoxicity. Because of the significant number of references in the literature to TH disruption as a principal mechanism of PCB developmental neurotoxicity, and the central relevance of the AhR to the mode of action of DL PCBs, these biological activities will also be briefly discussed in the context of their potential roles in mediating PCB developmental neurotoxicity.

### 5.1. Arylhydrocarbon Receptor (AhR) as a Molecular Target in PCB Developmental Neurotoxicity

AhR activation, which is the canonical mode of action for TCDD, is the distinguishing feature of DL PCBs. AhR-dependent signaling is implicated in the regulation of neurodevelopment in rodent models [166,167]. Altered patterns of neuronal cell proliferation, migration, and maturation have been observed in *Ahr*^-/-^ mice [168,169] and in mice genetically engineered to express constitutively active AhR [170]. These neurodevelopmental changes likely have functional consequences as evidenced by learning and memory deficits in *Ahr*^-/-^ mice [168]. Recent studies indicate that developmental exposure of mice to TCDD alters neuronal proliferation, migration, and dendritic arborization [167,168,170,171,172], and decreases USVs [173]; however, these neurotoxic effects are only observed at doses significantly higher than those required for induction of adverse outcomes in other organ systems. Nonetheless, these studies suggest that DL PCBs might interfere with typical neurodevelopment, as has been reported for other chemicals that activate the AhR, such as benzo[*a*]pyrene [174]. However, whether DL PCBs phenocopy the effects of TCCD on neurodevelopment at doses relevant to human exposure has yet to be determined.

Experimental evidence does suggest, however, that DL PCB interactions with the AhR may indirectly influence the developmental neurotoxicity of NDL PCBs. Four allelic variants of *Ahr* have been described in laboratory mice [175,176], and, of these, the higher-affinity variants (e.g., *AhR^b^* variants) appear to enhance susceptibility to the neurotoxic effects of PCBs via induction of cytochrome P450 (CYP) enzyme families 1A and 1B [176], which are believed to oxidize coplanar PCBs and non-coplanar PCBs, respectively [8]. This hypothesis is based on comparative studies of PCB developmental neurotoxicity in transgenic mice that express high-affinity AhR/wild-type CYP1A2 (*Ahr^b^/Cyp1a2^+/+^*) vs. mice nullizygous for *Cyp1a2* that express either the high- or low-affinity AhR (*Ahr^b^/Cyp1a2^-/-^* and *Ahr^d^/Cyp1a2^-/-^*, respectively) [177]. Gestational and lactational exposure to a mixture of DL (PCBs 77, 105, 118, 126, and 169) and NDL (PCBs 138, 153, and 180) congeners at 5.6 mg/kg/d in the maternal diet reduced serum thyroxine (T4) at PND 14 in offspring of all three genotypes. However, *Ahr^b^/Cyp1a2^-/-^* mice exhibited a greater T4 reduction than the other two genotypes [177]. Postnatal development of the cerebellum is particularly sensitive to TH insufficiency [175,176,177,178], and consistent with this, PCB-exposed *Ahr^b^/Cyp1a2^-/-^* mice exhibited defects in cerebellar structure at PND 25 [177]. In other studies by this group, exposure to this same PCB mixture on GD 10 and PND 5 resulted in learning and memory deficits in NOR and MWM tasks in both *Ahr^b^/Cyp1a2^-/-^* [178] and *Ahr^d^/Cyp1a2^-/-^* mice [179], enhanced startle response in *Ahr^d^/Cyp1a2^-/-^* mice [179], and increased locomotor activity in *Ahr^b^/Cyp1a2^-/-^* mice [178]. In contrast, *AhR^b^/Cyp1a2^+/+^* and *AhR^d^/Cyp1a2^+/+^* mice did not exhibit these neurobehavioral deficits. Collectively, these data suggest that activation of the AhR by DL PCBs indirectly modulates PCB developmental neurotoxicity via influences on PCB metabolism.

### 5.2. Thyroid Hormone-Mediated Mechanisms of PCB Developmental Neurotoxicity

TH influences many stages of neurodevelopment, and congenital hypothyroidism is linked to significant neurodevelopmental defects [180,181,182,183]. There is some evidence, albeit conflicting, that TH insufficiency may increase the risk of ASD [184,185] and is related to ADHD [186,187]. Numerous studies in experimental animal models have shown that developmental exposure to complex PCB mixtures decreases serum TH levels (reviewed in [153,154]); however, this relationship may be less consistent in humans [188,189]. Early animal studies supported a role for TH insufficiency in at least some of the auditory deficits induced by developmental Aroclor exposure [190,191]. However, the profile of cochlear damage caused by PCBs is not entirely consistent with other models of hypothyroidism, evidenced by the observation that TH replacement in Aroclor-exposed rats only partially ameliorated hearing deficits [153,190,191,192]. Additionally, later studies supported a TH-independent mechanism of PCB ototoxicity [193]. With regard to PCB effects on neurobehavior, data from animal models of developmental Aroclor exposure do not support a role for TH deficiency in Aroclor effects on learning and memory [47,194]. Similarly, studies with individual PCB congeners provide little support for the hypothesis that TH deficiency mediates the cognitive effects of developmental PCB exposures [155]. For example, NDL PCBs 28, 118, and 153 produce similar deficits in spatial learning and memory, but their effects on serum T4 levels vary from a marked reduction to no effect [101,195]. Conversely, DL PCBs 77 and 126 significantly reduce serum T4 [196], but result in few, if any, cognitive behavioral deficits [65,197,198].

In a typically developing brain, increased TH levels enhance and decreased TH levels attenuate dendritic arborization in cerebellar Purkinje cells [183,199]. However, maternal dietary exposure to Aroclor 1254 throughout gestation and lactation at doses of 1 or 6 mg/kg/d increased dendritic arborization or had no effect on dendritic arborization in Purkinje cells, respectively, even though both doses significantly decreased serum TH levels in dams and offspring [47,143]. Moreover, perinatal PCB 95 exposure significantly disrupts neuronal connectivity in the auditory cortex at a dose that has no quantifiable effect on auditory brainstem responses (ABRs) [200]. Since altered ABRs are a confirmed T4-dependent effect of PCBs [191], these observations suggest that T4-mediated mechanisms are not the major factor driving PCB effects on neuronal connectivity in the auditory cortex. Moreover, NDL PCB effects on dendritic growth are observed in serum-free cultures of primary hippocampal and cortical neurons, which are isolated from the hypothalamic-pituitary-thyroid axis and systemic TH influence [47,144,146,147,150].

While these observations argue against TH insufficiency playing a major role in PCB effects on learning and memory, they do not rule out the possibility that individual PCB congeners or their metabolites contribute to developmental neurotoxicity via direct actions at the level of the TH receptor (THR) on target cells in the brain. However, the data regarding the activity of PCBs and their metabolites at the THR or on THR-mediated gene transcription are inconsistent, with studies variably reporting agonistic effects, antagonistic interactions, or no activity [150,151,201,202]. A recent study that evaluated the THR activity of an NDL PCB mixture that proportionally mimics the PCB congener profile detected in the blood of pregnant women in Northern California observed neither agonistic nor antagonistic activity of these congeners when tested singly or in combination across a broad range of concentrations [69].

Collectively, the current literature does not provide strong support for the hypothesis that TH-dependent mechanisms play a predominant role in mediating PCB effects on neurodevelopment.

### 5.3. Calcium Signaling

Activity, or experience, is important in shaping neuronal architecture and is critical for refining neural circuits into functionally integrated networks [203,204,205,206,207]. Activity modulates not only structural aspects of neuronal connectivity, such as matching the number of neurons to the size of the target field and refining dendritic arbors and spine formation, but also neurochemical aspects of connectivity, such as neurotransmitter phenotype [208,209]. The effects of activity on neuronal connectivity are mediated primarily by modulation of intracellular Ca^2+^ [210,211], and precise control of intracellular Ca^2+^ dynamics is required for typical neurodevelopment [212,213,214]. Many candidate NDD risk genes encode proteins that generate intracellular Ca^2+^ signals or are themselves tightly regulated by local Ca^2+^ fluctuations [215,216]. These genes encode Ca^2+^ ion channels, neurotransmitter receptors, and Ca^2+^-regulated signaling proteins such as cAMP response element-binding protein (CREB) and Wnt2. Significant evidence supports the hypothesis that altered patterns of neuronal connectivity associated with NDDs are due in part to defects in neuronal Ca^2+^ signaling [210].

Many laboratories using diverse biophysical, biochemical, and cellular approaches have consistently shown that NDL PCBs, but not DL PCBs, alter Ca^2+^ dynamics in neurons, evidenced as increased intracellular Ca^2+^ levels and/or activation of Ca^2+^-dependent signaling events (reviewed in [55]). Mechanistic studies have identified multiple molecular mechanism(s) by which NDL PCBs influence intracellular Ca^2+^ levels in neurons. In vitro studies assessing pharmacological blockade of selected Ca^2+^ channels indicate that NDL PCBs increase extracellular Ca^2+^ influx via activation of l-type voltage-sensitive Ca^2+^ channels and NMDA receptors [217,218]. However, these effects are elicited only by high PCB concentrations (>10 µM) that also produce nonspecific changes in membrane fluidity [164]. NDL-PCBs also promote the release of Ca^2+^ from intracellular stores through sensitization of ryanodine receptors (RyR) [219,220,221] and inositol 1,4,5-trisphosphate receptors (IP_3_R) [222]. Of these, RyR sensitization is the most sensitive [55,219]. RyRs are microsomal Ca^2+^ ion channels that regulate endoplasmic reticulum storage and release of Ca^2+^. Picomolar to nanomolar concentrations of NDL PCBs interact directly with RyR channels to stabilize the channel in its open configuration [223,224]. This RYR-PCB interaction exhibits a stringent SAR, including stereoselectivity, as determined via biochemical, electrophysiological, cellular, and in vivo approaches [147,224,225,226,227,228,229,230].

RyR activity is implicated in the regulation of neuronal apoptosis [231,232] and dendritic growth [146,233], and several lines of evidence support a causal relationship between PCB effects on RyRs and PCB effects on neuronal apoptosis and dendritic growth. With regard to neuronal apoptosis, PCB 47, but neither PCB 77 nor PCB 104, triggers neuronal apoptosis in primary rat hippocampal neurons [121]. SAR studies demonstrate that PCB 47 is a RyR-active congener whereas PCB 77 and PCB 104 have negligible activity at the RyR [230]. Furthermore, the pro-apoptotic activity of PCB 47 in cultured neurons is inhibited by FLA 365, a selective RyR antagonist [234,235], but not by antagonists known to block PCB-mediated Ca^2+^ flux through l-type voltage-sensitive Ca^2+^ channels, NMDA receptors, or IP_3_Rs [121]. The signaling pathway(s) that connect RyR sensitization to neuronal apoptosis are not yet known, but presumably involve caspase-3-dependent [121], p53-independent [122] signaling mechanism(s).

SAR studies also support a causal role for RyR sensitization in PCB-induced dendritic growth. PCBs 95 and 136, but not PCB 66, increase dendritic arborization in primary rat hippocampal and cortical neurons [47,144,147]. PCBs 95 and 136 have potent RyR activity, whereas PCB 66 has negligible RyR activity [230]. PCB 136 is a chiral congener that atropselectively sensitizes RyRs [236], which translates into atropselective effects on dendritic growth. Specifically, the (−)-PCB 136 enantiomer potently sensitizes RyR and enhances dendritic growth, whereas the (+)-PCB 136 enantiomer lacks RyR activity and has no effect on dendritic growth [147]. Further supporting the hypothesis that RyR-dependent mechanisms underlie PCB effects on dendritic growth are observations that pharmacological blockade of RyRs inhibits the dendrite-promoting activity of PCB 95 and PCB 136 in primary rat hippocampal and cortical cultures [47,144,147], and siRNA knockdown of RyRs inhibits PCB 95-induced dendritic growth in rat dissociated hippocampal cell cultures and hippocampal slice cultures [144]. In vivo observations are consistent with a role for RyR in PCB developmental neurotoxicity. Aroclor 1254 is comprised predominantly of *ortho*-rich PCBs with significant RyR activity [219,220]. In rats, maternal dietary exposure to Aroclor 1254 throughout gestation and lactation showed that changes in dendritic growth and plasticity coincide with increased [^3^H]-ryanodine binding [47]. Since ryanodine only binds to RyR in its open conformation, an increase in ryanodine binding indicates increased RyR activity [55]. The dose-response relationship for the effects of Aroclor 1254 on dendritic growth and plasticity was similar to PCB effects on RyR expression, but not to PCB effects on TH levels or sex steroid-dependent developmental endpoints [47]. Increased RyR expression in the brain has also been associated with Aroclor 1254-induced changes in gene expression [237,238] and locomotor activity [239].

Ca^2+^-regulated translation- and transcription-dependent pathways regulate activity-dependent dendritic growth and spine formation [240,241,242]. PCB 95 promotes dendritic growth by engaging these same signaling pathways. Ca^2+^ imaging studies of primary rat hippocampal neurons demonstrate that RyR-active congeners, such as PCB 95 and the (−) enantiomer of PCB 136, increase the frequency and amplitude of Ca^2+^ oscillations, whereas congeners that are not RyR-active, such as PCB 66 or the (+) enantiomer of PCB 136, have no discernable effect on neuronal Ca^2+^ oscillations [146,147]. Pharmacological inhibition of RyRs blocks the intracellular Ca^2+^ oscillations triggered by PCB 95 and (−)-PCB 136, confirming that these PCB effects are RyR-dependent. In primary rat hippocampal neurons, the increase in intracellular Ca^2+^ caused by PCB 95 activates a Ca^2+^-dependent translational mechanism involving mechanistic target of rapamycin (mTOR) [145]. In addition, PCB 95 triggers sequential activation of CaMKK, CaMKIα/γ, MEK/ERK and CREB to increase transcription of Wnt2, which acts in an autocrine fashion to promote dendritic growth [146]. Pharmacological blockade of RyRs inhibits the activation of these signaling molecules, and experimental manipulations to inhibit the signaling molecules in these pathways effectively block PCB 95-induced dendritic growth [145,146]. Activation of CREB by PCB 95 also upregulates miR132, which suppresses the translation of p250GAP to promote synaptogenesis, evidenced by increased dendritic spine density and elevated frequency of miniature excitatory post-synaptic currents [149].

Collectively, the evidence available in the published literature provides strong support for the hypothesis that PCBs interfere with typical neurodevelopment via modulation of Ca^2+^-dependent signaling (Figure 2). These observations also suggest that that RyR-active NDL PCBs are significant contributors to PCB developmental neurotoxicity, a suggestion supported by recent epidemiological study designed to evaluate PCBs as risk factors for ASD [19]. The authors reported that while there were no significant associations for total PCBs and ASD, there were marginally significant associations linking DL-PCB exposure and reduced risk for diagnosis of non-typical development (adjusted OR: 0.41 (95% CI 0.15 to 1.14)) and between RyR-active NDL PCBs and increased risk for ASD diagnosis (adjusted OR: 2.63 (95% CI 0.87 to 7.97)). The authors of this study concluded “these analyses suggest the need to explore more deeply into subsets of PCBs as risk factors based on their function and structure in larger cohort studies where non-monotonic dose-response patterns can be better evaluated”.

### 5.4. Increased ROS as a Mechanism of PCB Developmental Neurotoxicity

Several in vitro studies have demonstrated that PCBs increase intracellular levels of ROS in neurons [243,244,245]. In cultured cerebellar granule neurons, exposure to Aroclor 1254, the NDL PCBs 4 or 153, or the hydroxylated metabolites of NDL PCBs 3, 14, 34, 35, 36, 39 or 68 increased intracellular ROS levels, which caused concentration-dependent cell death [243,245]. In contrast, DL PCBs had no effect on intracellular ROS in this model system. In vivo evidence of PCB-induced ROS in the developing brain is conflicting. In rats, gestational and lactational exposure to environmentally relevant levels of Aroclor 1254 (0.1 or 1.0 mg/kg/d) in the maternal diet resulted in increased oxidative stress biomarker expression across multiple brain regions [119]. In contrast, a different study of mice exposed via lactation to a mixture of six NDL PCBs at 1, 10 or 100 ng/kg for 14 days did not observe evidence of oxidative stress in the brain [246]. The discrepancy between these two studies may be due to species differences, timing and/or length of PCB exposure, or the assessment of different oxidative stress biomarkers, but more likely reflect differences in the dose and composition of PCB mixtures used in the different studies.

Increased levels of intracellular ROS are a significant trigger of pro-apoptotic signaling pathways [232,247,248,249], and several published studies suggest that PCB-induced ROS mediates the pro-apoptotic activity of NDL PCBs. In rats exposed throughout gestation to Aroclor 1254 in the maternal diet at 0.1 and 1 mg/kg/d, PCB-induced apoptosis coincided spatially and temporally with biomarkers of oxidative stress, specifically increased levels of 4-hydroxynonenal (4-HNE) and 3-nitrotyrosine (3-NT) as detected by western blotting [119]. A more direct link between PCB effects on intracellular ROS and neuronal apoptosis has been demonstrated in an in vitro model. Neuronal apoptosis induced by Aroclor 1254 or the NDL PCB 47 in primary rat hippocampal neurons was blocked by the antioxidant α-tocopherol. The observation that pharmacologic blockade of RyRs also blocked PCB effects on neuronal apoptosis [121] raises the possibility that RyRs and ROS interact to mediate PCB-induced apoptosis. For example, ROS may be a consequence of RyR activation. NDL PCBs stabilize RyRs in their open conformation, which allows release of Ca^2+^ from intracellular stores [55]. Increased intracellular Ca^2+^ can initiate apoptosis either by directly activating caspases and/or by increasing Ca^2+^ flux into mitochondria, the latter of which triggers production of ROS and results in release of cytochrome c from mitochondria with subsequent activation of caspases [115]. Alternatively, ROS may initiate apoptosis via targeted interactions with the RyR to activate pro-apoptotic signaling [247,250]. ROS have been shown to interact directly with RyR cysteine residues to heighten the probability of channel opening [251,252,253,254]. Thus, NDL PCBs may sensitize RyRs indirectly as a consequence of PCB-induced increase in ROS [121]. In support of this alternative model, quinone-generated ROS have been shown to enhance RyR Ca^2+^ conductance in rabbit sarcoplasmic reticulum membranes [251]. RyR channels are tightly regulated by changes in redox potential [254,255] and are very sensitive to modification by redox-active toxicants [251], providing a plausible mechanistic link between Ca^2+^ dysregulation and oxidative stress. As recently noted [14], it may also be that PCBs independently influence RyR activation and ROS generation, and that each of these effects augments the other in a feed-forward mechanism.

Collectively, the available data, while limited, support the hypothesis that increased intracellular ROS contribute to PCB developmental neurotoxicity, specifically PCB-induced neuronal apoptosis, potentially via RyR-dependent mechanisms (Figure 2). Additional studies are needed to confirm and extend these observations to additional PCB congeners. There is also a significant data gap as to the significance of ROS in mediating PCB developmental neurotoxicity in vivo.

## 6. Emerging Evidence that Non-Legacy LC-PCBs Interfere with Typical Neurodevelopment

Over the past decade, PCB congeners not found in the mixtures synthesized prior to the ban on commercial PCB production are increasingly being identified in the human chemosphere [6,69,256]. Most of these are LC-PCBs (reviewed in [8]). The non-legacy LC-PCB, PCB 11, has recently been detected in the serum of dairy animals [257] and in commercial cow’s milk in Northern California [258]. Of even greater concern, PCB 11 has also been found in serum of pregnant women living in Northern California [13,69] as well as women and their adolescent children living in the greater Chicago area and rural Iowa [6]. Analysis of PCBs in the serum of pregnant women at increased risk for having a child with an NDD revealed that the non-legacy LC-PCBs 28 and 11 were the two most abundant PCB congeners, and together they comprised almost 75% of the PCB burden in the maternal serum samples [69].

In contrast to the legacy PCBs, particularly the legacy HC-PCBs, relatively little is known about the potential developmental neurotoxicity of the non-legacy LC-PCBs. An early study reported that the hydroxylated metabolites of eight different LC-PCBs increase ROS formation and induce cell death in cultured cerebellar granule cells [245]. Later studies demonstrated that PCB 11 and its known human hydroxyl and sulfated metabolites increase dendritic arborization in primary rat hippocampal and cortical neurons [259]. These dendritic effects are observed in cultures exposed to PCB 11 at fM concentrations, which are within the range of PCB 11 concentrations detected in the serum of pregnant women [259]. In contrast to the HC-PCBs 95 and 136, PCB 11 and its metabolites not only increase dendritic arborization, but also enhance axonal growth [13]. Subsequent mechanistic studies indicated that the dendrite-promoting activity of PCB 11 is not blocked by pharmacological antagonism of the RyR, the AhR, or the THR [259]. The antioxidant α-tocopherol and inhibitors of the L-type Ca^2+^ channel or IP_3_R also have no effect on PCB 11-induced dendritic growth [259]. However, pharmacological inhibition or siRNA knockdown of CREB effectively blocks the dendritic effects of PCB 11 [259]. PCB 95, an NDL legacy congener, also enhances dendritic arborization through CREB activation in vitro [146], suggesting that CREB may be a point of mechanistic convergence for PCB-induced dendritic growth. However, the upstream molecular targets of PCB 95 vs. PCB 11 differ, with PCB 95 activating CREB via RyR sensitization [144] and PCB 11 activating CREB through RyR-independent mechanism(s) (Figure 2). The proximal target(s) through which PCB 11 triggers CREB-dependent dendritic growth have yet to be identified. Also not yet known is whether developmental exposure to PCB 11, or other non-legacy LC-PCBs, affects dendritic arborization in vivo and, if so, whether dendritic effects coincide with neurobehavioral deficits.

## 7. Relevance of PCB Developmental Neurotoxicity to NDDs

Genetic, histologic, in vivo imaging and functional data in the human literature all point to altered patterns of neuronal connectivity as the biological basis for intellectual disabilities and the behavioral and cognitive abnormalities associated with many NDDs, including ASD and ADHD [107,112,132,133]. The candidate genes most strongly associated with NDD etiology encode proteins that regulate the organizational aspects of neuronal network patterning during development and influence the balance of excitatory to inhibitory synapses [104,107,109,260,261]. Enlarged dendritic arbors and increased dendritic spine density are observed post mortem in the hippocampus of individuals with ASD [262], and imaging studies of individuals with ASD and ADHD have revealed evidence of disrupted regional connectivity within the brain [81,263,264,265,266,267]. For example, ASD is hypothesized to reflect cortical dysconnectivity characterized by local hyperconnectivity and long-range regional hypoconnectivity [133,262,268,269].

Epidemiological studies have identified an association between developmental PCB exposure and increased risk of ASD [18,19,20,21,22,23] and ADHD [24,25,26,27], with a recent study suggesting that RyR-active NDL PCBs may be driving this association [19]. Observations that PCBs, and in particular NDL congeners, modulate the same neurodevelopmental processes that are altered in ASD, ADHD and intellectual disabilities, suggest a biologically plausible mechanism to explain these associations. However, it is not likely that developmental PCB exposures cause NDDs, but rather that PCBs interact with genetic susceptibility factors to influence individual NDD risk [104]. Genetically determined imbalances in synaptic connectivity may provide a biological substrate for elevated susceptibility to PCBs effects on neuronal apoptosis, axonal and dendritic growth, and activity-dependent refinement of synaptic connections [104,261].

One way that heritable genetic susceptibilities might amplify the adverse effects of PCB exposure is if both factors (genes × environment) converge to dysregulate the same signaling pathways that control neuronal connectivity during critical periods of development [104,270] (Figure 3). As discussed earlier (Section 4 and Section 5; also reviewed in [14]), NDL PCBs promote neuronal apoptosis and enhance dendritic arborization by activating Ca^2+^-dependent signaling. A large number of candidate NDD risk genes encode proteins involved in Ca^2+^ signaling [215,216], and defects in neuronal Ca^2+^ signaling contribute to altered patterns of neuronal connectivity associated with NDDs [210]. Studies investigating gain-of-function human *RYR* mutations have demonstrated that specific *RYR* mutations confer sex-, gene-, and dose-dependent vulnerability to pharmacological (halogenated anesthetic) and environmental (heat) stressors that trigger malignant hyperthermia and subsequent muscle damage in otherwise asymptomatic individuals [271,272]. Moreover, PCB 95 is significantly more potent and efficacious in causing Ca^2+^ dyshomeostasis in human mutant R615C-*RYR1* channels compared to wild type *RYR1* channels in vitro [273]. While a previous SNP study concluded there was no evidence of an association between *RYR3* and ASD in a Japanese population [274], a subsequent GWAS identified *RYR2* as an ASD candidate gene when evaluating sex as a covariate [275].

Directly relevant to the proposal that PCBs converge on signaling pathways that are dysregulated by NDD risk genes, the effects of PCB 95 and PCB 11 map onto Ca^2+^-dependent signaling pathways disrupted in NDDs. For example, Timothy syndrome is caused by a gain-of-function mutation in the L-type Ca^2+^ channel CaV1.2. This syndrome has a 100% incidence of NDD and 60% rate of ASD [276]. Neurons differentiated from induced pluripotent stem cells derived from Timothy syndrome patients exhibit increased intracellular Ca^2+^ oscillations and increased expression of genes associated with Ca^2+^-dependent regulation of CREB, including CaMK [277]. Altered CREB signaling is implicated in various NDDs [278,279,280,281]. Consistent with these clinical observations, PI3Kγ knockout mice exhibit an ADHD-like behavioral phenotype coincident with enhanced CREB signaling [279] and transgenic mice expressing human mutations in CREB binding protein exhibit increased stereotypy, social deficits, and learning and memory deficits [282]. One downstream effector of CREB activation is miR132, which has been shown to be elevated in individuals with ASD [283,284,285]. Wnt, another downstream effector of CREB, is also implicated in the pathogenesis of ASD, and is thought to underlie the stereotyped, repetitive behaviors observed in ASD patients [286,287]. There is also evidence suggesting that disrupted Wnt signaling can impair social behaviors as a result of disrupted brain organizational patterning and interhemispheric connectivity in a manner reminiscent of ASD [288].

Whether PCBs interact with heritable mutations in Ca^2+^ signaling to increase risk of adverse neurodevelopmental outcomes has yet to be determined, but published observations suggest hypotheses to be tested in both epidemiological and experimental animal studies.

## 8. Conclusions, Data Gaps, and Directions for Future Study

The available neurobehavioral and mechanistic studies in experimental animal and cell culture models suggest that NDL PCBs are driving PCB developmental neurotoxicity. This conclusion is supported by a recent epidemiological investigation of PCBs as risk factors for ASD that found no association between developmental exposure to DL PCBs and neuropsychiatric deficits in children, but did identify an association, albeit weak, between NDL PCBs, specifically those that are RyR-active, and increased risk of ASD diagnosis [19]. While the data currently available do not support a major role for DL PCBs in PCB developmental neurotoxicity, they also do not rule out the possibility that DL PCBs play some role, possibly in modifying neurotoxic responses to NDL PCBs [177,179]. However, clearly there is a need for more studies evaluating potential neurodevelopmental effects of individual PCB congeners and mixtures that better replicate contemporary human exposures than the legacy Aroclor mixtures. This will require a more comprehensive understanding of the congener profile and concentrations of contemporary human maternal and neonatal PCB exposures. To achieve this, it will be necessary to move away from the traditional practice of quantifying legacy indicator PCBs and towards non-biased screening for a broad spectrum of PCBs and their metabolites, including non-legacy PCBs, in relevant human tissue samples.

There still remains uncertainty as to the molecular mechanisms mediating PCB effects on neurodevelopment. Mechanistic studies suggest that NDL PCBs cause neurobehavioral deficits by interfering with neuronal apoptosis and dendritic arborization, in large part via RyR, ROS and CREB-dependent mechanisms. However, much of the literature describing mechanisms of PCB developmental neurotoxicity is based on either complex mixtures or a small subset of individual congeners, leaving significant uncertainty regarding the generalizability of these mechanisms to the broad spectrum of PCBs. In addition to determining congener-specificity, there is a need to identify dose-response relationships of PCB effects on molecular and cellular endpoints, and to establish causal relationships between molecular, cellular and neurobehavioral endpoints with the goal of identifying robust adverse outcome pathways linking molecular initiating events to neurotoxic outcomes at the organism and population levels [289]. The continual evolution of high(er)-throughput methfods for analyzing molecular and cellular endpoints will be of great utility in addressing these data gaps.

To date, there is a paucity of data—both epidemiological and nonclinical—describing the developmental neurotoxicity of the non-legacy LC-PCBs. The urgency of addressing this data gap is suggested by evidence that these congeners represent the most abundant PCBs in the serum of pregnant women at increased risk of having a child with an NDD [69]. This is heightened by experimental data demonstrating that PCB 11 and human relevant metabolites of PCB 11 alter neuronal morphogenesis in vitro at extremely low concentrations (reviewed in [67]). It needs to be determined whether these in vitro effects of PCB 11 are predictive of in vivo effects, and whether they extend to other LC-PCBs. This point is underscored by a recent risk assessment that detected several airborne LC-PCBs for which there are limited to no toxicological data [290].

In summary, while PCBs are arguably among the most studied developmental neurotoxicants, there remain significant, and important, questions that require investment of research resources in order to rigorously assess the risk that these contaminants pose to the developing human brain.

## Figures and Tables

**Figure 1 ijms-21-01013-f001:**
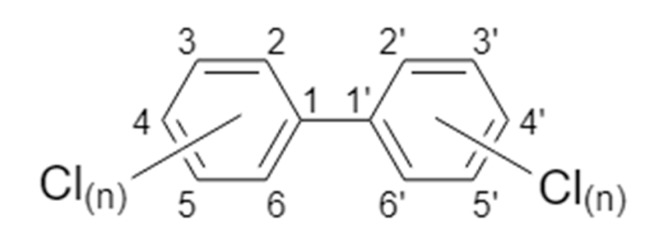
Basic PCB chemical structure. DL PCB congeners are typically *meta* (3, 3′, 5, 5′) and *para* (4, 4′) substituted, with no chlorines at the *ortho* positions (2, 2′, 6, 6′), and are coplanar. NDL PCBs typically have more than one *ortho*-substituted chlorine and adopt a non-coplanar structure.

**Figure 2 ijms-21-01013-f002:**
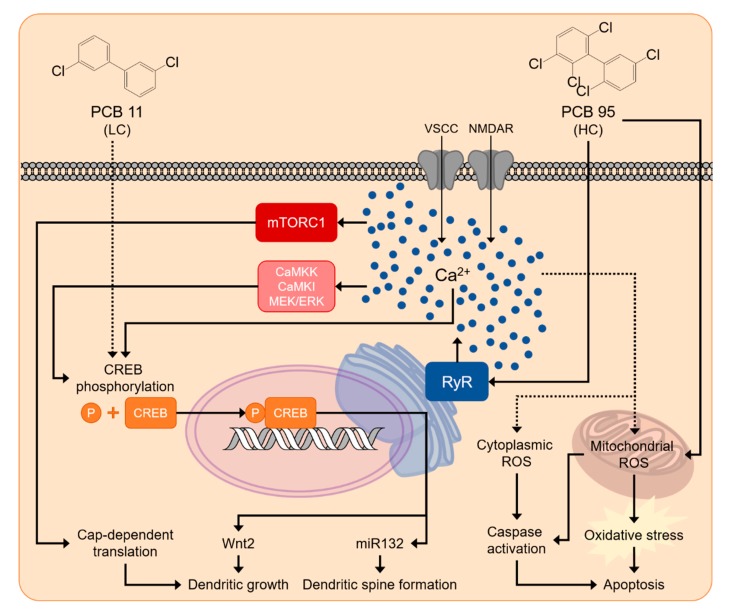
Experimental evidence supports a mechanistic model in which NDL PCBs alter neuronal morphogenesis and promote neuronal apoptosis via Ca^2+^-dependent and/or ROS-dependent mechanisms. CaMK, Ca^2+^/calmodulin-dependent protein kinase; CREB, cAMP response element-binding protein; HC, highly chlorinated; LC, lightly chlorinated; MEK/ERK, mitogen-activated protein kinase kinase/extracellular signal-regulated kinase; NMDAR, NMDA receptor; RyR, ryanodine receptor; VSCC, voltage-sensitive calcium channel. Solid lines indicate that experimental evidence directly links the upstream and downstream event; in contrast, dotted lines indicate a link but the intervening steps have yet to be identified.

**Figure 3 ijms-21-01013-f003:**
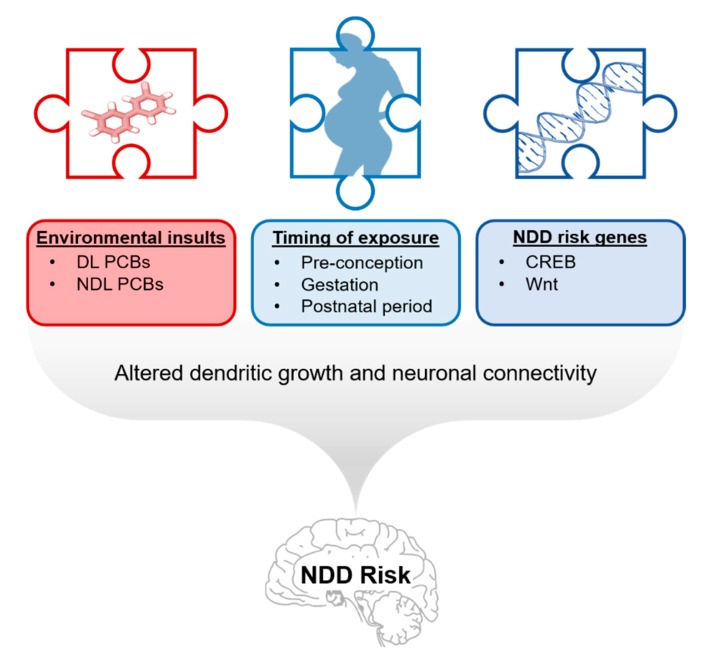
PCBs may interact with heritable mutations in cAMP response element-binding protein (CREB) signaling to influence neurodevelopmental disorder (NDD) risk.

**Table 1 ijms-21-01013-t001:** Effects of developmental PCB exposure on locomotor behavior.

Model	Exposure	Dose(s)	Route of Exposure	Exposure Window	Findings	Ref.
Mouse (ICR)	A1254	6 mg/kg/d, 18 mg/kg/d	Gavage	Lactation (PND 7–21), Postnatal (PND 22–42)	↑ Locomotor activity in females	[74]
Mouse (ICR)	A1254	18 mg/kg/d	Injection (i.p.)	Prenatal (GD 6-PND 0), Lactation (PND 0–21)	↑ Locomotor activity	[75]
Mouse (Swiss albino)	NDL PCB mixture (PCBs 28, 52, 101, 138, 153, 180)	1 ng/kg/d, 10 ng/kg/d, 100 ng/kg/d	Gavage	Lactation (PND 0–21)	↓ Locomotor activity in 1 and 10 ng/kg males	[76]
Rat (Sprague-Dawley)	A1221	0.5 mg/kg, 1 mg/kg	Injection (i.p.)	Prenatal (GD 16, 18)	↑ Distance traveled in LDB in male offspring	[77]
Rat (Wistar)	PCB 52, PCB 138, PCB 180	1 mg/kg/d	Dietary (jelly)	Prenatal (GD 7-PND 0), Lactation (PND 0–21)	↓ Locomotor activity in PCB 138 and PCB 180 groups in males; ↓ Locomotor activity in PCB 138 group in females	[78]
Rat (Wistar)	PCB 52, PCB 138, PCB 180	1 mg/kg/d	Dietary (jelly)	Prenatal (GD 7-PND 0), Lactation (PND 0–21)	↓ Time spent on rotarod in PCB 52 group for both sexes	[79]

“Postnatal” exposure indicates PCB(s) were given directly to the pup, while “lactation” exposure indicates PCB(s) were administered directly to the dam and indirectly to the pup via consumption of milk. Abbreviations: A1221 = Aroclor 1221; A1254 = Aroclor 1254; i.p. = intraperitoneal; GD = gestational day; LDB = light-dark box; PND = postnatal day; ↑ = increased; ↓ = decreased.

**Table 2 ijms-21-01013-t002:** Effects of developmental PCB exposure on social behaviors.

Model	Exposure	Dose(s)	Route of Exposure	Exposure Window	Findings	Ref.
Mouse (CD1)	NDL PCB mixture (PCBs 28, 52, 101, 138, 153, 180)	10 ng/kg/d, 1 µg/kg/d	Dietary (chow)	Prenatal (GD 6-PND 0), Lactation (PND 0–21)	↑ Sociability in early adulthood in both sexes; ↑ Social novelty in both sexes (1 µg/kg/d); ↓ Social interaction in middle-aged males (1 µg/kg/d)	[89]
Rat (Sprague-Dawley)	A1221	1 mg/kg	Injection (i.p.)	Prenatal (GD 16, 18, 20), Postnatal ^1^ (PND 24, 26, 28)	↑ USV calls in female (prenatal and postnatal); ↑ Affiliative wrestling behavior in female (prenatal and postnatal); ↓ Time spent near novel animal of opposite sex in males exposed postnatally	[90]
Rat (Sprague-Dawley)	A1221	0.5 mg/kg, 1 mg/kg	Injection (i.p.)	Prenatal (GD 16, 18)	↓ Social interaction with novel conspecific in 0.5 mg/kg males	[91]
Rat (Sprague-Dawley)	A1221	0.5 mg/kg/d, 1 mg/kg/d	Injection (i.p.)	Prenatal (GD 16, 18)	↑ USV calls in males in sociosexual context; No effect on sociosexual behavior in females; ↓ Sociosexual interaction (nose-nose sniffs) in males	[92]
Rat (Sprague-Dawley)	Mixture (PCBs 47, 77)	12.5 mg/kg/d, 25 mg/kg/d	Dietary (chow)	Prenatal (GD 0-PND 0), Lactation (PND 0–21)	↓ Social recognition in 25 mg/kg males; ↓ Social investigation in males (both doses)	[93]

^1^ “Postnatal” exposure indicates PCB(s) were given directly to the pup, while “lactation” exposure indicates PCB(s) were administered directly to the dam and indirectly to the pup via consumption of milk. Abbreviations: A1221 = Aroclor 1221; i.p. = intraperitoneal; GD = gestational day; PND = postnatal day; USV = ultrasonic vocalization; ↑ = increased; ↓ = decreased.

**Table 3 ijms-21-01013-t003:** Effects of developmental PCB exposure on cognitive behavior and executive function.

Model	Exposure	Dose(s)	Route of Exposure	Exposure Window	Findings	Ref.
Mouse (ICR)	A1254	6 mg/kg/d, 8 mg/kg/d	Gavage	Lactation (PND 7–21), Juvenile (PND 22–42)	↓ NOR performance in females; No effect on alternation behavior in either sex	[74]
Mouse (ICR)	A1254	18 mg/kg/d	Injection (i.p.)	Prenatal (GD 6-PND 0), Lactation (PND 0–21), Postnatal (PND 21–35)	↓ Object-based attention in NOR task; ↓ Latency to enter open arms of EPM; ↑ Time spent in open arms of EPM	[75]
Mouse (Swiss albino)	NDL PCB mixture (PCBs 28, 52, 101, 138, 153, 180)	1 ng/kg/d, 10 ng/kg/d, 100 ng/kg/d	Gavage	Lactation (PND 0–21)	↑ Escape latency in water escape task in males (1 and 100 ng/kg), ↑ Anxiety-like behavior in EPM and LDB tasks (1 and 10 ng/kg), No effect on performance in MWM task	[76]
Mouse (Swiss albino)	NDL PCB mixture (PCBs 28, 52, 101, 138, 153, 180)	10 ng/kg/d	Gavage	Lactation (PND 0–21)	No effect on short-term memory in spontaneous alternation task; No effect on learning acquisition in MWM task	[97]
Rat (Sprague-Dawley)	A1221	1 mg/kg/d	Injection (i.p.)	Prenatal (GD 16, 18, 20), Postnatal ^1^ (PND 24, 26, 28)	↑ Entries into and time spent in open arms of EPM in females during both periods	[90]
Rat (Long-Evans)	Fox River	3 mg/kg/d, 6 mg/kg/d	Dietary (cookie)	Pre-conception (28 d), Prenatal (GD 0-PND 0), Lactation (PND 0–21)	↓ DRL performance in 3 mg/kg females; ↓ Inhibitory control in both sexes (both doses)	[98]
Rat (Long-Evans)	Fox River	3 mg/kg/d, 6 mg/kg/d	Dietary (cookie)	Postnatal ^1^ (PND 27–50)	↑ Response latency in cue discrimination phase of set-shifting task in males (3 mg/kg/d); ↓ Errors to criterion in position reversal in males (both doses); ↓ Perseverative errors in position reversal in males (both doses); No effect on DRL performance	[99]
Rat (Wistar)	PCB 52, PCB 138, PCB 180	1 mg/kg/d	Dietary (jelly)	Prenatal (GD 7-PND 0), Lactation (PND 0–21)	↓ Learning in Y maze visual discrimination task for PCB 138 and 180 in both sexes	[79]

^1^ “Postnatal” exposure indicates PCB(s) were given directly to the pup, while “lactation” exposure indicates PCB(s) were administered directly to the dam and indirectly to the pup via consumption of milk. Abbreviations: A1221 = Aroclor 1221; A1254 = Aroclor 1254; i.p. = intraperitoneal; GD = gestational day; PND = postnatal day; EPM = elevated plus maze; DRL = differential reward of low-rate; LDB = light-dark box; MWM = Morris water maze; NOR = novel object recognition; ↑ = increased; ↓ = decreased.

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
