# Peer review of "Evidence Implicating Non-Dioxin-Like Congeners as the Key Mediators of Polychlorinated Biphenyl (PCB) Developmental Neurotoxicity"

_ijms, 2020, doi:10.3390/ijms21031013_

Round 1

Reviewer 1 Report

The authors reviewed last ten years literature on PCB-induced developmental neurotoxicity. The first part of the review the authors distinguisc between neurobehavioral studies on hyperactivity, social deficit and cognitive disfunction. In the second part of the review the focus on the molecular basis of PCB-driven neurotoxicity.

The review is well written and complete, anyway I have some suggestions.

-I would suggest to resume the concept of section 4 and 5 in tables (with the relative references) as done for the content of section 3. This is of great help for the reader in order to make easier comparisons.

-For the contents of section 6 and 7 I would suggest the possibility of evaluating a picture descibing the most relevant molecular pathways invoked in the text.

Author Response

Reviewer 1

The authors reviewed last ten years literature on PCB-induced developmental neurotoxicity. The first part of the review the authors distinguish between neurobehavioral studies on hyperactivity, social deficit and cognitive dysfunction. In the second part of the review the focus on the molecular basis of PCB-driven neurotoxicity. The review is well written and complete, anyway I have some suggestions.

We sincerely thank the reviewer for the positive feedback.

I would suggest to resume the concept of section 4 and 5 in tables (with the relative references) as done for the content of section 3. This is of great help for the reader in order to make easier comparisons.

We have carefully considered this recommendation, and while we appreciate the merit of the idea, we do not think it feasible given that we discuss approximately 80 studies in Sections 4 and 5. Summarizing 80 studies in tables as we did with the much more limited literature summarized in Section 3 would not be possible in the time frame we were given for returning the revised manuscript to the journal, and it would add significantly to the length of an already long manuscript.

For the contents of section 6 and 7, I would suggest the possibility of evaluating a picture describing the most relevant molecular pathways invoked in the text.

We agree that the addition of a mechanistic figure would add clarity. We have developed a new figure that is included in the revised manuscript.  

Reviewer 2 Report

SUMMARY: The authors have completed a comprehensive review of recent data regarding the developmental neurotoxicity of polychlorinated biphenyls (PCBs). They clearly articulate multiple proposed mechanisms of toxicity and focus on the emerging concerns about lightly chlorinated PCB congeners. The review is well written and organized, but a few suggestions are offered to improve clarity and depth.

INTRODUCTION: Some data on total production and estimates of total PCBs remaining in the environment would help demonstrate the ongoing magnitude of the concern.

Line 47: Can the authors provide specific data on amounts found in human tissue and environmental samples?

Section 2. Lines 104-109. Although the authors’ focus is appropriately on the most recent literature, it might be helpful to reference a review that summarizes the earlier reports of differential effects of individual PCBs and various mixtures. Intriguingly, there have been occasional reports of improved function following PCB exposure. See suggested references.

Tables. It would be helpful and more informative if the strain of mouse and rat are reported, since there can be significant differences in genetics and metabolism. Could this account for some of the variability seen across studies?

Table 3. It’s not clear why tests of anxiety-like behavior (elevated plus maze) are included in this section.

Table 3. Dopamine is mentioned in the table legend, but not in the main table. Is some data missing?

Line 285. Minor grammatical correction is needed. “to determine whether there the…”

Section 4.1 and tables. The authors report the greatest effects in the cerebellum, but present no data from motor function experiments. Is this a knowledge gap that should be addressed? Could these effects have a different mechanism of action (e.g. thyroid hormone depletion)?

Section 5.1 There is abundant literature demonstrating that developmental exposure to AHR agonists result in neurotoxic effects. Elaborating on the known effects of benzo[a]pyrene and other polycyclic aromatic hydrocarbons that activate the AHR would help balance the overall discussion. Could there be a cumulative effect from co-exposure to PCB mixtures that exacerbates the toxicity? Or mitigates it through increased metabolism and clearance? Please also see suggested references for recent data suggesting the AHR might also have a role in the toxicity of some lightly chlorinated PCBs of current interest.

Section 5.3. An illustration of the proposed major mechanism of action would be helpful.

Suggested references:

Ulbrich and Stahlmann. 2004. Developmental toxicity of polychlorinated biphenyls (PCBs): a systematic review of experimental data. Arch Toxicol 78: 252–268

Takeuchi et al. 2017. Effects of unintentional PCBs in pigments and chemical products on transcriptional activity via aryl hydrocarbon and nuclear hormone receptors. Environmental Pollution 227 306e313

Author Response

Reviewer 2

SUMMARY: The authors have completed a comprehensive review of recent data regarding the developmental neurotoxicity of polychlorinated biphenyls (PCBs). They clearly articulate multiple proposed mechanisms of toxicity and focus on the emerging concerns about lightly chlorinated PCB congeners. The review is well written and organized, but a few suggestions are offered to improve clarity and depth.

We sincerely thank the reviewer for the positive feedback.

INTRODUCTION: Some data on total production and estimates of total PCBs remaining in the environment would help demonstrate the ongoing magnitude of the concern.

We have added the requested information to the first paragraph of the Introduction (lines 32-33 and lines 42-43).

Line 47: Can the authors provide specific data on amounts found in human tissue and environmental samples? This information has been added to the Introduction (lines 49-51).

Section 2. Lines 104-109. Although the authors’ focus is appropriately on the most recent literature, it might be helpful to reference a review that summarizes the earlier reports of differential effects of individual PCBs and various mixtures. Intriguingly, there have been occasional reports of improved function following PCB exposure. See suggested references.

We thank the reviewer for providing references to consider including in our review. We cite 2 reviews that cover the older literature looking at general cognitive function and executive following developmental PCB exposure (references 46-47) as well as a recent review detailing older and newer literature about the social deficits following PCB exposure (reference 45). The Ulbrich and Stahlmann paper suggested by the reviewer is another older review that primarily focuses on molecular and mechanistic effects with a brief discussion of PCB behavioral effects. It contains some outdated information regarding behavioral study design (e.g., the authors endorse the use of multiple pups per litter for behavioral studies, which is not a practice recommended by the USEPA, OECD or neurobehavioral experts). However, given that it does cover the older literature for developmental PCB exposure and locomotor behavior, we have included it as an additional reference in the manuscript. The Takeuchi et al. paper is not applicable to our manuscript on PCB neurotoxicity as this study uses an ovarian-derived cell line and the observed effects might not be relevant to the developing brain.

Tables. It would be helpful and more informative if the strain of mouse and rat are reported, since there can be significant differences in genetics and metabolism. Could this account for some of the variability seen across studies?

We agree that this would provide a useful additional point of comparison. The strains have been added to all tables.

Table 3. It’s not clear why tests of anxiety-like behavior (elevated plus maze) are included in this section.

We chose to include EPM studies in Table 3 given that this test assesses anxiety-like behavior in a non-social context without the anxiogenic sensory stimuli present in other tests (e.g., bright light in a light-dark box test). EPM “relies on the inherent conflict between exploration of a novel area and avoidance of its aversive features” [1]. In humans, anxiety is comorbid with ASD at an estimated rate of 30-50% and it is thought to influence hallmark ASD symptoms, including cognitive inflexibility that is associated with restricted behaviors and aversion to novelty [2]. Thus, the reluctance of an animal to explore the open arms of an EPM may in fact be related to cognitive inflexibility as a result of fear/uncertainty about the open arms. In addition, EPM is arguably not the sole test of anxiety-like behavior reviewed in our manuscript. ASD is also characterized by deficits in social interaction and communication, and the presence of comorbid social anxiety can enhance the severity of these deficits. In nonclinical animal models, this can be tested in a variety of social behavior assays, such as social novelty [1], which we discuss in Section 3.2/Table 2. Taken together, the current literature suggests that comorbid anxiety disorders and ASD have significant symptom overlap [3-5]. Therefore, in an effort to keep the behavior literature tables as streamlined as possible, we chose to include the tests for anxiety-associated behaviors nested under the broader behavioral domains.

Table 3. Dopamine is mentioned in the table legend, but not in the main table. Is some data missing?

Thank you for pointing out this oversight on our part. This was left in the legend in error from a previous iteration of the table. It has been deleted from the revised manuscript.

Line 285. Minor grammatical correction is needed. “to determine whether there the…”

We have corrected this grammatical error.

Section 4.1 and tables. The authors report the greatest effects in the cerebellum, but present no data from motor function experiments. Is this a knowledge gap that should be addressed? Could these effects have a different mechanism of action (e.g. thyroid hormone depletion)?

Table 1 reports the recent evidence of the effects of developmental PCB exposure on altered locomotor behavior and rotarod performance, which reflect motor function. The cerebellar apoptosis study discussed in Section 4.1 [6] did not address behavioral deficits, though it is possible that effects on motor function might result from increased apoptosis in this brain region. As discussed Section 3.1/Table 1, other rat studies have reported reduced locomotor activity and rotarod performance in the context of a similar developmental Aroclor 1254 exposure. Recent studies have also identified that the cerebellum is important in higher-order cognitive and emotional functions [7] and that this region is implicated in ASD etiology [8,9]. Therefore, it is possible that increased apoptosis due to developmental PCB exposure could contribute to both cognitive deficits and motor effects, but, to our knowledge, no study has looked at both of these endpoints in the context of PCB exposure.

Section 5.1 There is abundant literature demonstrating that developmental exposure to AHR agonists result in neurotoxic effects. Elaborating on the known effects of benzo[a]pyrene and other polycyclic aromatic hydrocarbons that activate the AHR would help balance the overall discussion. Could there be a cumulative effect from co-exposure to PCB mixtures that exacerbates the toxicity? Or mitigates it through increased metabolism and clearance? Please also see suggested references for recent data suggesting the AHR might also have a role in the toxicity of some lightly chlorinated PCBs of current interest.

We again thank the reviewer for providing references for this point. We agree that co-exposure to other toxicants like BaP and PAHs (among many others) is likely and that these chemicals could potentially modulate the effects of PCB exposure or vice versa. While, we feel that a detailed discussion of the developmental neurotoxicity of these particular toxicant classes is beyond of the scope of this manuscript, we have included reference to a review of the developmental neurotoxicity of benzo[a]pyrene in section 5.1. We respectfully have decided to not reference the Takeuchi et al. paper because it is not relevant to developmental neurotoxicity. While the effects of lightly chlorinated PCBs on the AhR are certainly of interest, the effects of AhR activation are extremely dependent on cell type and PCB congener. For example, a previous study from our lab found that lightly chlorinated PCB 11 had no AhR activity in a pituitary tumor-derived cell line [10], which differs from the Takeuchi study that found agonistic AhR activity of lightly chlorinated PCBs 1, 35, and 56. We also found that AhR activity was not involved in the effects of PCB 11 on dendritic growth [10]. 

Section 5.3. An illustration of the proposed major mechanism of action would be helpful.

We agree that a summary figure would be useful and have added one to the revised manuscript.

References cited in the response to Reviewers:

File, S.E.; Lippa, A.S.; Beer, B.; Lippa, M.T. Animal tests of anxiety. Curr. Protoc. Neurosci. 2004, Chapter 8, Unit 8 3, doi:10.1002/0471142301.ns0803s26. Lai, M.C.; Lombardo, M.V.; Baron-Cohen, S. Autism. Lancet 2014, 383, 896-910, doi:10.1016/S0140-6736(13)61539-1. Zaboski, B.A.; Storch, E.A. Comorbid autism spectrum disorder and anxiety disorders: a brief review. Future Neurol. 2018, 13, 31-37, doi:10.2217/fnl-2017-0030. Lau, B.Y.; Leong, R.; Uljarevic, M.; Lerh, J.W.; Rodgers, J.; Hollocks, M.J.; South, M.; McConachie, H.; Ozsivadjian, A.; Van Hecke, A., et al. Anxiety in young people with autism spectrum disorder: Common and autism-related anxiety experiences and their associations with individual characteristics. Autism 2019, 10.1177/1362361319886246, 1362361319886246, doi:10.1177/1362361319886246. South, M.; Rodgers, J. Sensory, Emotional and Cognitive Contributions to Anxiety in Autism Spectrum Disorders. Front. Hum. Neurosci. 2017, 11, 20, doi:10.3389/fnhum.2017.00020. Yang, D.; Lein, P.J. Polychlorinated biphenyls increase apoptosis in the developing rat brain. Curr Neurobiol 2010, 1, 70-76. Koziol, L.F.; Budding, D.; Andreasen, N.; D'Arrigo, S.; Bulgheroni, S.; Imamizu, H.; Ito, M.; Manto, M.; Marvel, C.; Parker, K., et al. Consensus paper: the cerebellum's role in movement and cognition. Cerebellum 2014, 13, 151-177, doi:10.1007/s12311-013-0511-x. Becker, E.B.; Stoodley, C.J. Autism spectrum disorder and the cerebellum. Int. Rev. Neurobiol. 2013, 113, 1-34, doi:10.1016/B978-0-12-418700-9.00001-0. Fatemi, S.H. Cerebellum and autism. Cerebellum 2013, 12, 778-779, doi:10.1007/s12311-013-0484-9. Sethi, S.; Keil, K.P.; Lein, P.J. 3,3'-Dichlorobiphenyl (PCB 11) promotes dendritic arborization in primary rat cortical neurons via a CREB-dependent mechanism. Archives of toxicology 2018, 92, 3337-3345, doi:10.1007/s00204-018-2307-8.